



# SIBaR: A New Method for Background Quantification and Removal from Mobile Air Pollution Measurements

Blake Actkinson[1], Katherine Ensor[2], Robert J. Griffin[1,3]

[1]Department of Civil and Environmental Engineering, Rice University, Houston, TX 77005, USA
5 [2]Department of Statistics, Rice University, Houston, TX 77005, USA
[3]Department of Chemical and Biomolecular Engineering, Rice University, Houston, TX 77005, USA

*Correspondence to*: Robert Griffin (rob.griffin@rice.edu)

**Abstract.** Mobile monitoring is becoming increasingly popular for characterizing air pollution on fine spatial scales. In identifying local source contributions to measured pollutant concentrations, the detection and quantification of background are 10 key steps in many mobile monitoring studies, but the methodology to do so requires further development to improve replicability. Here we discuss a new method for quantifying and removing background in mobile monitoring studies, State Informed Background Removal (SIBaR). The method employs Hidden Markov Models (HMMs), a popular modelling technique that detects regime changes in time series. We discuss the development of SIBaR and assess its performance on an external dataset. We find 86% agreement between the predictions made by SIBaR and the predetermined allocation of 15 background and non-background data points. We compare five-minute averages of SIBaR-derived background $NO_x$ measurements to five-minute averages of $NO_x$ measurements taken by a stationary monitor sitting 70 m above ground level near downtown Houston, finding greater disagreement between SIBaR and the stationary monitor than the disagreement between other background detection techniques and the same stationary monitor. We then assess its application to a data set collected in Houston, TX, by mapping the fraction of points designated as background and comparing source contributions to 20 those derived using other published background detection and removal techniques. Results suggest that SIBaR could serve as a framework for improved background quantification and removal in future mobile monitoring studies.

## 1 Introduction

Understanding air pollution exposure is important, as it has been linked to various adverse health conditions (Caplin et al., 2019; Zhang et al., 2018). Mobile monitoring, a technique in which continuous air pollution measurements are collected using 25 instrumentation on a mobile platform, is becoming increasingly important for characterizing exposure because air pollution varies on spatial scales finer than the typical distance between stationary monitors (Apte et al., 2017; Chambliss et al., 2020; Messier et al., 2018).

A key component of mobile monitoring analysis is identifying the background, defined here as measured air pollution 30 independent of local source influences (Brantley et al., 2014). Background quantification is vital from both policy and exposure





perspectives, as it is important to assess the contribution of local sources to pollution concentrations accurately. Table 1 summarizes the wide variety of methods used to estimate background in studies incorporating mobile monitoring published within the past five years. The wide variance in the approaches used is problematic, as estimates of source contributions to measurements have been shown to be sensitive to the technique used (Brantley et al., 2014). To improve the replicability and

power of mobile monitoring studies, a more consistent technique for background estimation is needed.

Designing a method to determine background in mobile monitoring studies presents several challenges. Measurements in remote locations are often regarded as the most reliable representation of background; however, remote locations may be inaccessible for some mobile monitoring studies and are themselves subject to occasional source influences. These drawbacks

make time series methods for determining background more desirable. However, time series-based methods often rely on setting static time windows, which are usually determined by the expected duration of influence from source plumes within the mobile monitoring study (Bukowiecki et al., 2002). The underlying physical representation of time series methods remains unclear for more extensive mobile monitoring campaigns, as the setting of static time windows does not often capture the entire variation in time scales that source impacts can have on mobile measurements.


Here we show the results of a newly developed method called **S**tate-**I**nformed **Ba**ckground **R**emoval (SIBaR) used to estimate background for several traffic related air pollutants. The method incorporates Hidden Markov Models (HMMs), a time series regime modelling technique used in a wide variety of contexts in signals processing, finance, and the social sciences and which has been used to model background in stationary monitors (Gómez-Losada et al., 2016, 2018, 2019; Visser and Speekenbrink,

2010). HMMs assume that observations within a time series are drawn from probability distributions governed by a hidden sequence of states. We propose decoding this hidden sequence of states as a way to determine whether measurements were taken in locations representative of background versus locations subject to local influences. We illustrate that a more physically meaningful representation of background is captured in this modelling context for mobile monitoring time series and show its application to a wide variety of traffic related air pollutant measurements. As a proof of concept, we run the method on a

published external dataset already marked as background and non-background and assess its performance, and we compare a SIBaR-derived nitrogen oxide ($NO_x$) background signal with stationary rooftop monitor $NO_x$ measurements. As a first application, we map points binned as background by SIBaR to show their spatial distributions. As a proof of importance, we highlight differences in mapped source contributions derived from SIBaR background and background derived from other time series-based techniques. Results indicate that our consistent method for background identification and removal has

significant impact on mapped mobile source contributions.





| Study | Method Used to Determine Background Concentration |
|---|---|
| Apte et al., 2017 | Applied 10-s moving average filter, then selected the smaller of the given data value or the 2-min $5^{th}$ percentile to derive baseline concentrations. |
| Brantley et al., 2019 | Fitted quantile regression with cubic natural spline basis expansion of time with degrees of freedom equal to the number of hours in the time series. |
| Hankey and Marshall, 2015 | Used pollutant-specific underwrite functions to estimate instantaneous background concentrations and subtracted these concentrations from the original time series, averaged reference monitor measurements, then added averaged measurements to underwrite adjusted time series. |
| Hankey et al., 2019 | Used hourly averaged measurements in centrally located site for additive correction factor; used daily median fixed-site measurement for temporal correction factor. |
| Hudda et al., 2014 | Applied rolling 30-s $5^{th}$ percentile of the original time series. |
| Larson et al., 2017 | Applied 10-min rolling minimum. |
| Li et al., 2019 | Applied 1-min moving median filter, then calculated 1-hr rolling $5^{th}$ percentile of smoothed data; additionally, used wavelet decomposition to isolate concentration changes across 8 hours at stationary monitors, then subtracted lowest decoupled concentration from mobile monitoring time series across 15-min time windows. |
| Patton et al., 2014 | Used mobile measurements in designated urban background neighborhoods removed from highway. |
| Robinson et al., 2018 | Linearly interpolated averaged data collected at designated background locations. |
| Shairsingh et al., 2018 | Applied rolling 60-s mean, then applied spline of minimums technique (Brantley et al., 2014) across different time windows dependent on a desired background scale. |
| Tessum et al., 2018 | Used daily $5^{th}$ percentile for all pollutants other than fine particle number concentration; used rolling 30-min $5^{th}$ percentile for fine particle number concentration. |
| Van den Bossche et al., 2015 | Used averaged measurements from stationary monitor located in an urban green to apply additive correction factors to measurements greater than background then averaged site measurement and multiplicative correction factors to measurements lower than background. |

**Table 1. Summary of Previous Methodologies for Estimating Background Levels of Air Pollution in Mobile Monitoring Campaigns.**



## 2 Methods

### 2.1 Mobile Campaign

Measurements were taken during the Houston Mobile Monitoring Google Street View (GSV) campaign and are described in detail elsewhere (Miller et al., 2020). In brief, for a nine-month campaign, instruments were loaded into two gasoline-powered GSV cars that sampled every drivable road in twenty-two different census tracts in the greater Houston area. The time of day and day of week for each census tract visit were determined to minimize temporal biases to the greatest extent possible. Census tracts are included in the current analysis if they were sampled a minimum of fifteen times during this nine-month period (Apte et al., 2017; Li et al., 2019). Details and names used to describe each of the census tracts considered are given in Table S1 in the Supplement. Individual observations are aggregated to 50-meter points in neighborhood and 90-meter points on highway using a road network created from U.S. Census TigerLine roads (TIGER/Line Shapefile, 2018). More details on the road network creation and data quality control are provided elsewhere (Miller et al., 2020). The pollutants measured were black carbon (BC), carbon dioxide ($CO_2$), nitric oxide (NO), nitrogen dioxide ($NO_2$) ($NO_x = NO + NO_2$), ozone ($O_3$), fine particulate matter ($PM_{2.5}$), and ultrafine particle (UFP) number concentration. In this analysis, $PM_{2.5}$ (predominantly secondary), $O_3$ (purely secondary), and UFP (somewhat secondary) are not considered. Instruments used are described in Table S2.

### 2.2 Hidden Markov Model Categorization – The Background Partitioning Step

Because HMM fits are sensitive to outliers in the time series that often can be attributed to the noise of the instrument, we smooth each pollutant time series with a moving average time window of thirty seconds, then log transform the resulting smoothed time series. Time series observations are segregated by day and for each car separately, and HMMs are fit to each day's worth of data. The HMMs attempt to maximize the log-likelihood, $L_C$, determined by the sum of the forward variables $\alpha_T(i)$:

$$L_c = \sum_i^N \alpha_T(i) \tag{1}$$

in which $i$ designates state $i$ (total states $N$) at the last realization of the time series $T$. The forward variables are derived recursively as:

$$\alpha_1(i) = \pi_i p(y_1|\theta_i, z) \tag{2}$$

$$\alpha_{t+1}(j) = \sum_i^N \big(\alpha_t(i)a_{ij}\big)p(y_t|\theta_j, z) \tag{3}$$

in which $\pi_i$ represents the initial probability for state $i$, $a_{ij}$ represents the state transition probability from state $i$ to state $j$, and $p(y_t|\theta_i, z)$ represents the conditional probability of observation $y_t$ conditioned on the parameters $\theta_i$ governed by state $i$ and



any additional covariates $z$. For the purposes of our work, we parametrize time linearly and include it as an additional covariate to capture temporal variations in background. We also assume that the probability distributions governing $y_t$ are log normal.

The log-likelihood of equation (1) is maximized using the expectation maximization algorithm (Dempster et al., 1977; Visser
and Speekenbrink, 2010). Initial starting values of the transition probabilities are bootstrapped 150 times to produce 150 candidate models because convergence to a maximum likelihood can be affected by the starting values. The model with the greatest log-likelihood is then selected for decoding via the Viterbi algorithm (Forney, 1973). The Viterbi algorithm seeks to maximize the joint probability of both observations and state sequence $(q_1, \dots, q_T)$ given the parameters. We define a variable $\delta$ recursively as


$$\delta_{t+1}(j) = \left[ max\ \delta_t\ (i) a_{ij} \right] p(y_{t+1} | \theta_j, z) \tag{4}$$

with the initialization

$$\delta_1(i) = \pi_i p(y_1 | \theta_i, z) \tag{5}$$

To retrieve the state sequence, we create a matrix $\psi$ such that

$$\psi_1(i) = 0 \qquad 1 \leq i \leq N \tag{6}$$

$$\psi_t(j) = argmax\big(\delta_{t-1}(i) a_{ij}\big) \qquad 1 \leq j \leq N, 2 \leq t \leq T \tag{7}$$

We retrieve the state sequence by backtracking:

$$q_T = argmax[\delta_T(i)] \qquad 1 \leq i \leq N \tag{8}$$

$$q_t = \psi_{t+1}(q_{t+1}) \qquad t = T-1,\ T-2, \dots 1 \tag{9}$$

This state sequence is then used to designate points as background or source. State assigned points with the lower median are
designated background. An example of a decoded sequence is given in Figure 1 for $NO_x$ (after retransformation).





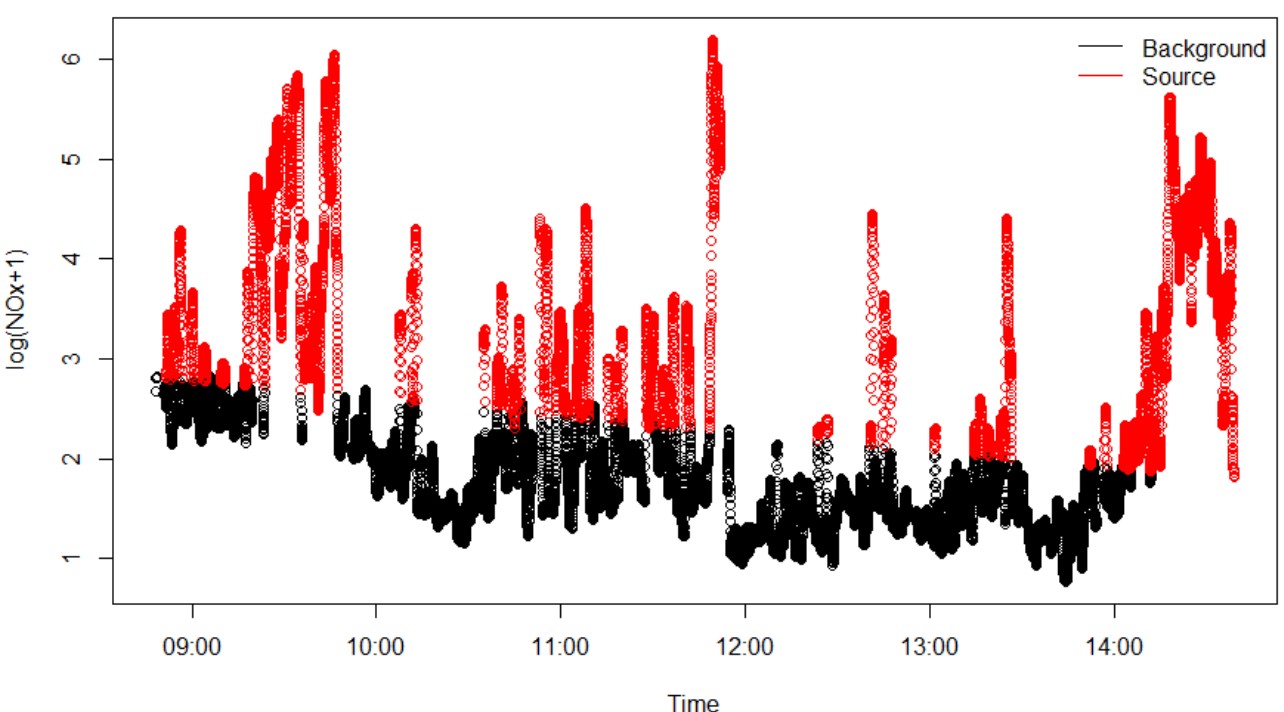

**Figure 1. Example of decoded state sequence for log transformed NOₓ which has been retransformed. Source designated points are red, and background designated points are black.**

### 2.3 2D Thin Plate Spline Fit

After HMMs have been fit to all time series data, all background designated points throughout the mobile monitoring campaign are compiled and fit to a two-dimensional (2D) thin plate spline as a function of time and day expressed as a tensor product.

The 2D splines are fit using the R package mgcv with k = 5 (Wood, 2003). We select a 2D spline fit to all background points overall instead of fitting splines day by day to prevent extrapolation in instances where the first measurements taken are categorized as source. Relative maximum likelihood is used to determine the smoothing parameters of the spline. The result is a day-to-day spline that represents the background across the sampling campaign for each pollutant. A depiction of the background spline for log transformed NOₓ is given in Figure 2.




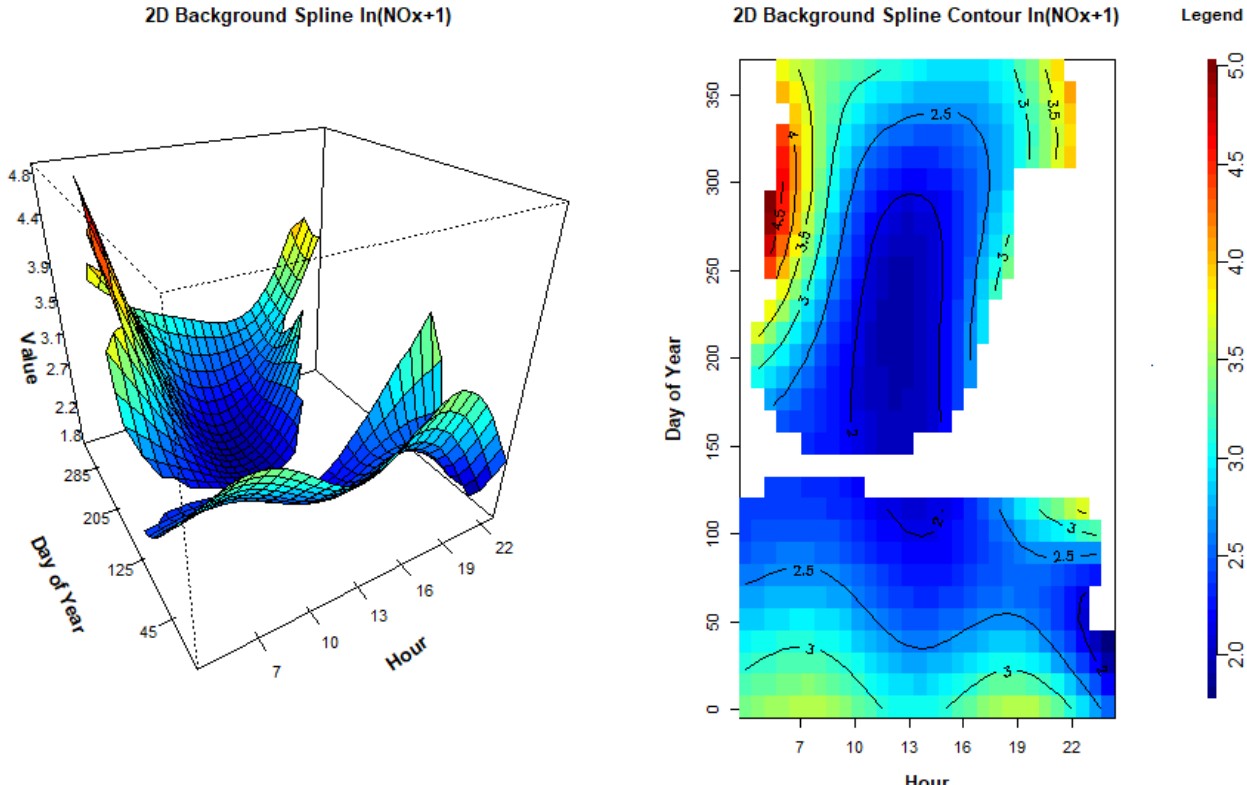

**Figure 2. (left) Depiction of 2D background spline for log transformed NO$_x$. Hour of the day is depicted on the axis going across the page, day of the year depicted on this axis going into the page. (right) The same data shown as a contour plot. The same color scale is used in both panels.**

Because SIBaR's partitioning step periodically generates background assigned points that differ from one another for the same time series, we perform a test to evaluate its robustness. We run SIBaR 25 times and evaluate the pairwise root-mean-square error (RMSE, defined in (10)) between each set of generated background predictions. The pairwise RMSE values for the first twelve runs are given in Table S3. We calculate an average RMSE of 0.01 between each background signal and conclude that

the fitting step is robust to small changes in background assigned points in the partitioning step.

### 2.4 Validating the Partitioning Step on an External Dataset and Stationary Monitor Comparison

To determine the validity of the partitioning step, we test it on a dataset published in Brantley et al. (2014). In that study, a van taking mobile measurements of CO systematically looped a route in which it drove through a predefined background location, on transects to a highway, and on the highway itself (Brantley et al., 2014). The measurements taken in the prescribed

background location were marked as background, and all other measurements were marked as non-background. We run the





partitioning step on these data to determine how well SIBaR captures the measurements taken in the background location of the study.

We also compare the background derived by SIBaR to five-minute averages of a stationary monitor located in Houston. The
monitor is stationed on top of Moody Tower on the University of Houston campus located between downtown Houston and the Houston Ship Channel. The site is seventy meters above ground level and has been used as an indicator of city-wide emission patterns in previous studies (Lefer et al., 2010; Luke et al., 2010). In this work, because of its elevation we assume it to be the stationary monitor most indicative of trends in Houston background $NO_x$. To put these comparisons in context with previously published work, we repeat the same process using background derived from a moving two-minute fifth-percentile
baseline ("Apte," Apte et al., 2017) and from a tenth quantile regression onto a cubic spline basis expansion with the degrees of freedom equal to the number of hours in the time series ("Brantley," Brantley et al., 2019).

**2.5 Generating Mapped Fractional Background Contribution and Source Contribution Maps**

We explore the spatial extent of our HMM decoded categorizations from the partitioning step by creating mapped fractional background contribution maps. After aggregating time series observations to road segment points created within our road
segment network, we sum the number of observations designated as the background state and divide by the total number of observations assigned to that road segment point. We map the results and present them in Section 4.1.

In section 4.2, we derive source contributions (source signal = original signal – background signal) using our background method and map them. To derive our source contributions, we make predictions for the background for each time series
observation collected using the derived background spline and then subtract those predictions from the original time series observations. We also derive source contributions using the Apte and Brantley techniques. We create the maps using the same methodology as Miller et al. (2020), described briefly here. Using our created road segment network, we take the mean of measurements the car makes as it drives past a road segment point in our network, coined the drive pass mean. We take the median of these drive pass means and map the result. To prevent the temporal conditions of drive pass means occurring within
four hours of one another from biasing the overall median of the sample, we take the median of drive pass means occurring within that four hour time window to generate a four-hour drive pass median and then take the median of all four-hour drive pass medians to derive the map reduced median. We perform this procedure for the source contributions derived using our method and the source contributions derived using the other published methods.



## 3 Results – Proof of Concept

### 3.1 Background Partitioning on an External Dataset

A comparison between SIBaR's partitioning and the partitioning originally published by Brantley et al. (2014) is given in Figure 3. Following the steps in SIBaR, the data are first smoothed with a thirty-second time window to dampen the influence of outliers. The HMM fitting step is performed, and the resulting state sequence decoded. The percentage of matching background/non-background designations is computed. The SIBaR partitioning step is able to match 86% of the originally

published background/non-background designations. The mismatches could be attributed to the transition between the background/non-background portions of the route in the original study, which is observed in Figure 3 in the periods where background points show larger values than source points near periods of the transition (for example., the last blue spike at approximately 8:45AM). Mismatches also could be a result of the effects of traffic on measurements in the background designated portion of the route and the inability of the SIBaR linearity assumption to capture all fine scale temporal variations

within the background (see equations (2)-(3)).



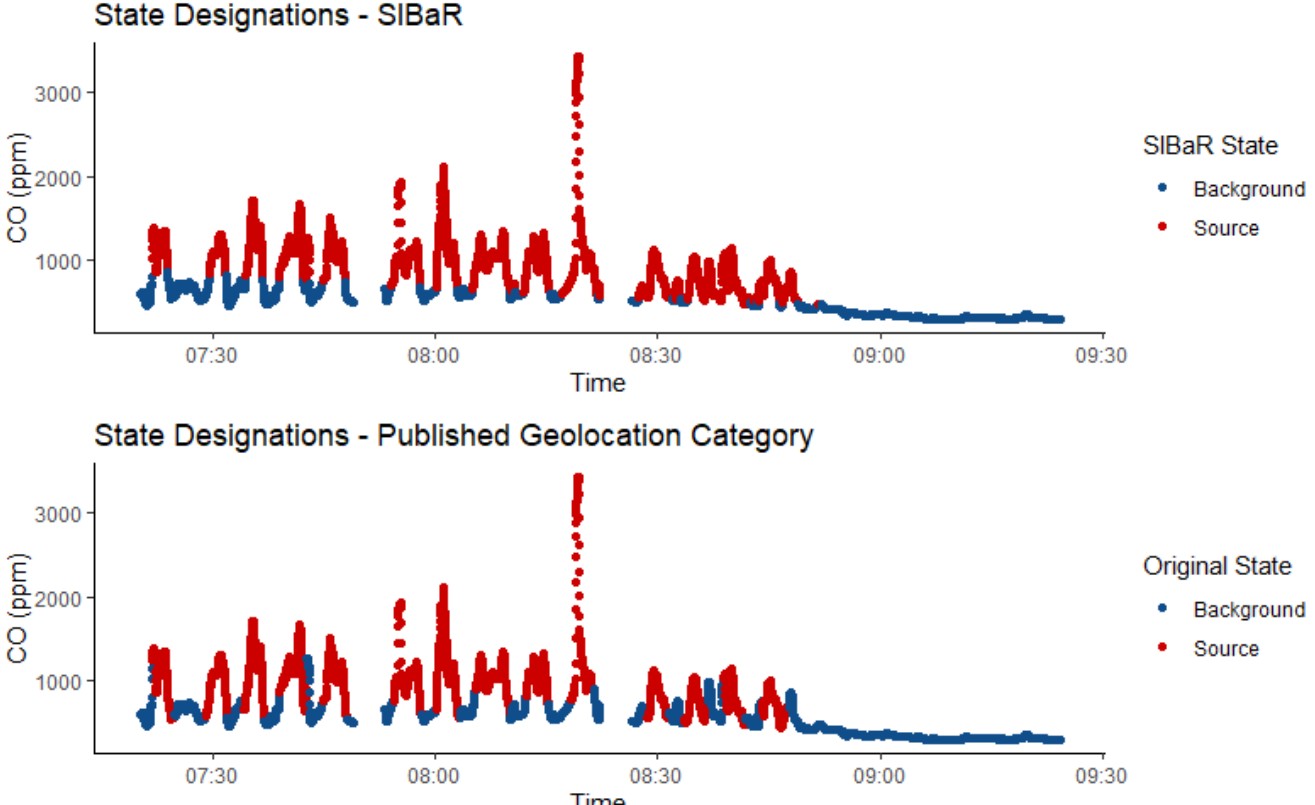

**Figure 3. Comparison between SIBaR-predicted background and source states and the originally published designation from Brantley et al. (2014).**

In running this test, we note that the method is sensitive to the smoothing time window used. Figure S1 in the supplement illustrates SIBaR predictions for three different smoothing time windows on the same CO data set and shows that the method produces different state categorizations depending on the window used. We hypothesize that, in this instance, smoothing

reduces the skewness of the data such that it better fits two switched lognormal Gaussian distributions. Different time windows should be investigated in using this method. In this instance, we use the thirty-second time window because background designated points are lower than their source designated counterparts.

### 3.2 Application to Stationary Urban Background NOx Measurements

Five-minute forward averages of SIBaR background predictions for $NO_x$ from GSV measurements (irrespective of location)

are taken and compared to five-minute averages of the Moody Tower $NO_x$ measurements on the same day. Only five-minute averages with complete data throughout the time interval are utilized. We filter measurements such that they fall between 10 AM and 4 PM local time to remove any potential influences of rush hour traffic on the stationary monitor. We compute the





root mean square error (RMSE) and mean absolute error (MAE) between SIBaR's five-minute background averages and the monitor's five minute averages, defined below:


$$RMSE = \sqrt{\frac{\sum_i^T (\hat{n}_i - n_i)^2}{T}} \qquad (10)$$

$$MAE = \frac{\sum_i^T |\hat{n}_i - n_i|}{T} \qquad (11)$$

in which $\hat{n}_i$ is the SIBaR-estimated five-minute average, $n_i$ is the monitor five-minute average, $i$ is an index which describes a matching five-minute time stamp, and $T$ is the total number of time stamps. We repeat the process for both the Apte and Brantley techniques and tabulate the values of RMSE and MAE to assess SIBaR performance relative to what has been published in the literature. The average $NO_x$ measurement reported by the stationary monitor during the time period is 10.34 ppb. RMSE and MAE values for all three techniques are given in Table 2.


| Technique | RMSE | MAE |
|-----------|------|-----|
| Apte | 10.98 ppb | 5.98 ppb |
| Brantley | 7.48 ppb | 4.08 ppb |
| SIBaR | 11.61 ppb | 7.77 ppb |


Table 2. Metrics that describe differences mapped source contributions between the three techniques for $NO_x$.

The Brantley technique consistently out performs the other two techniques in having lower RMSE and MAE values. In the sixty days of data that we examined, we find that the Brantley technique has the lowest RMSE values fifty out of sixty days 220 tested, with SIBaR having the lowest RMSE value six out of the sixty days and the Apte technique having the remaining four.





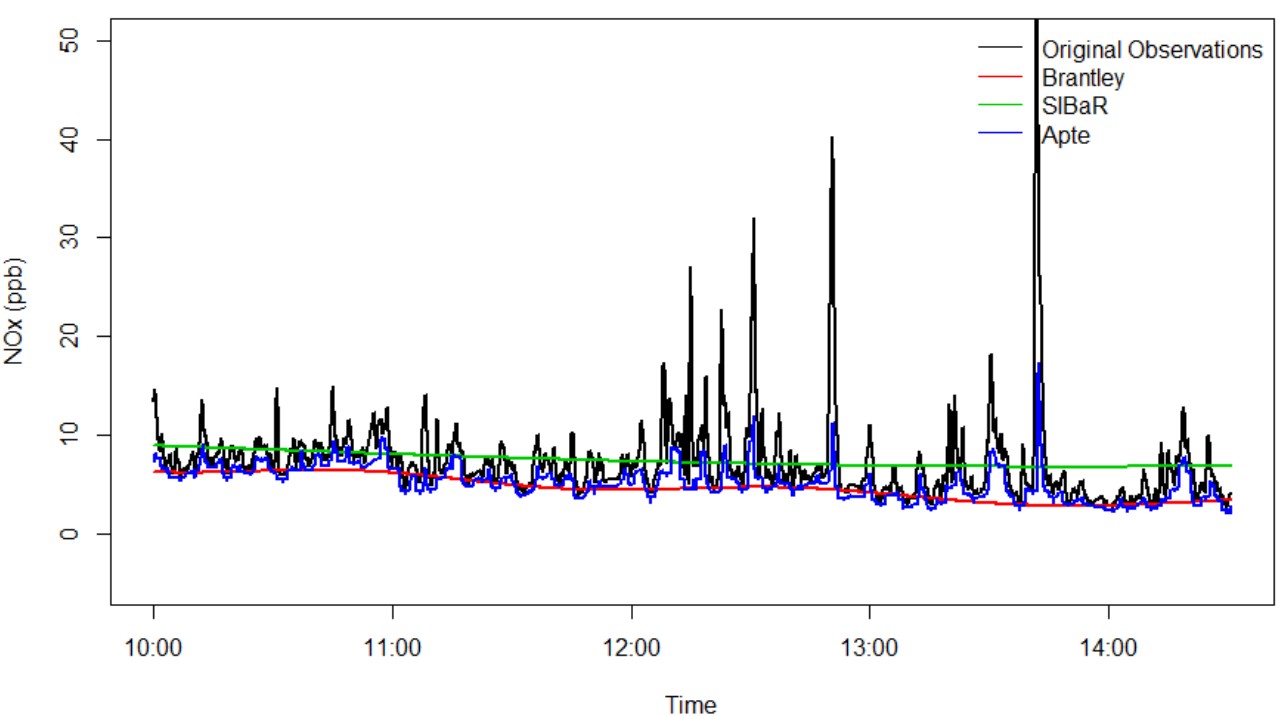

**Figure 4. Example time series of background juxtaposed with the original mobile monitoring time series observations.**

To illustrate differences in flexibility between the three background techniques, an example daily time series of each background technique's predictions is plotted in Figure 4. The Brantley technique's background signal is more flexible than SIBaR's. SIBaR's flexibility is computationally limited due to the sheer number of points fit, necessitated by the fact that background assigned points on different days are needed to prevent extrapolation on days in which the first points in the time series are source designated. Greater flexibility could allow background estimates to better capture temporal variations in temporal background compared to less flexible techniques. However, the reverse could also be true: by being too flexible, the Brantley background technique could be capturing local pollution influences which coincide with local pollution influences in the stationary monitor. While the monitor's measurements have been used as an indication of urban wide emission patterns, it does sit within a mile of a highway and rail line, potentially subjecting it to localized source influences. More work is needed to determine which of these outcomes is more likely to be the case.



## 4 Results – Preliminary Applications

### 4.1 Mapped Fractional Background State Contributions

For the Houston mobile campaign, maps detailing the fractional contribution of the background state to the overall mapped points are created for $CO_2$ and $NO_x$. Individual observations assigned to a road segment point have their category designations

assigned to the same point. The number of observations assigned the background category are then divided by the total number of observations assigned to the point to determine the fractional background state contribution. Figure 5 shows these census tract maps for $NO_x$. Figure S2 in the supplement shows the maps for $CO_2$. It is important to note that these maps represent the fraction of the measurements that are categorized as background or source for the given pollutant at a given location.

We note the following about the broad spatial patterns in mapped background state fraction presented in Figure 5. First, background state designated points dominate residential areas for both pollutants. This is encouraging, as it is expected that few point sources of these two pollutants would be found in residential neighborhoods except for those near industrial activity (Miller et al., 2020). Second, source state designated points dominate highways and busy arterials, which is expected given the large amounts of traffic on these roads. Finally, we note the appearance of source-dominated hotspots in front of point sources

identified in our previous work (Miller et al., 2020). This is encouraging given that we found these road segments to be elevated compared to their surrounding neighbourhood domain.

We take the background state fractions depicted in Figure 5 and bin them by distance to highway. The results are presented in Figure 6. We do the same for $CO_2$ and present the results in Figure S3 in the supplement. The exponential behaviour exhibited

in Figure 6 mirrors published exponential decays in roadside source pollutant concentrations (Apte et al., 2017; Karner et al., 2010), while the sizeable interquartile ranges within each bin highlight the complexity and variability of source roadside gradients, which depend on emission rates, meteorology, geography, and other factors (Baldwin et al., 2015; Patton et al., 2014).





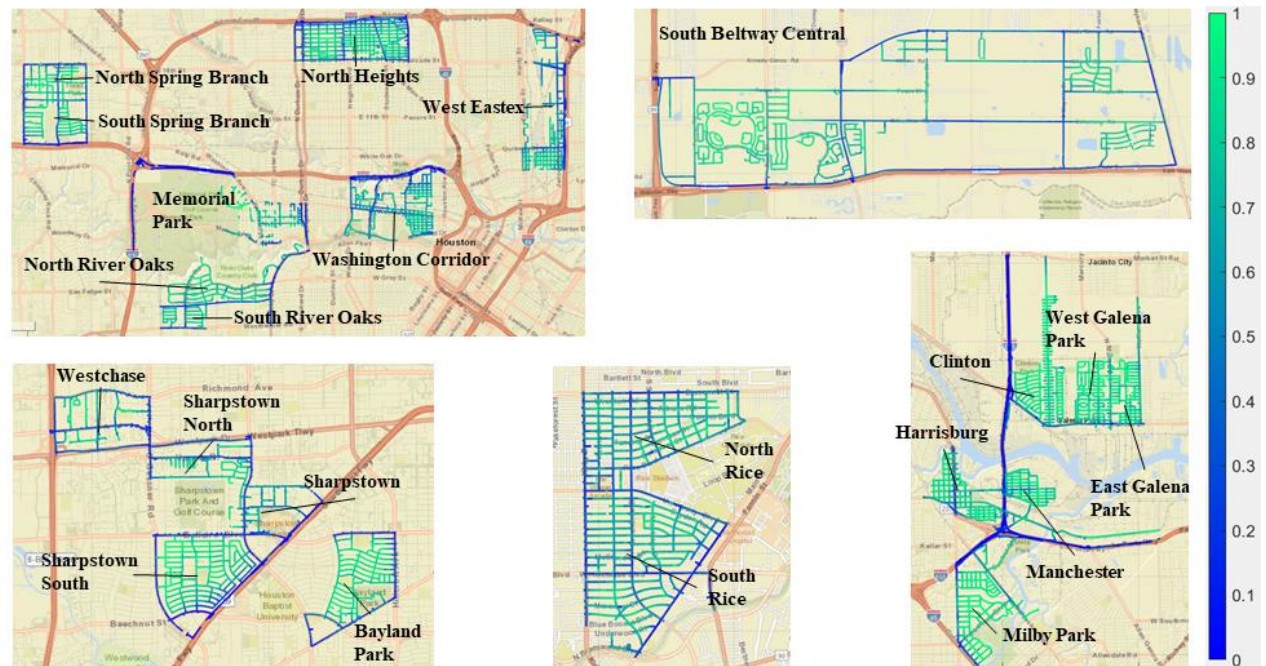

**Figure 5. Fraction of points aggregated to road segment network designated as background in SIBaR decoded states for NOₓ. Maps were generated following the methods outlined in Section 2.4. Points are mapped on a scale of 0 to 1; 1 implies all points aggregated to that road segment were designated as background, 0 implies all points were designated as non-background. Details of the census tracts are provided in Table S1. Basemap generated by Matlab geobasemap 'streets' and is hosted by ESRI (Sources: Esri, DeLorme, HERE, USGS, Intermap, iPC, NRCAN, Esri Japan, METI, Esri China (Hong Kong), Esri (Thailand), MapmyIndia, Tomtom).**







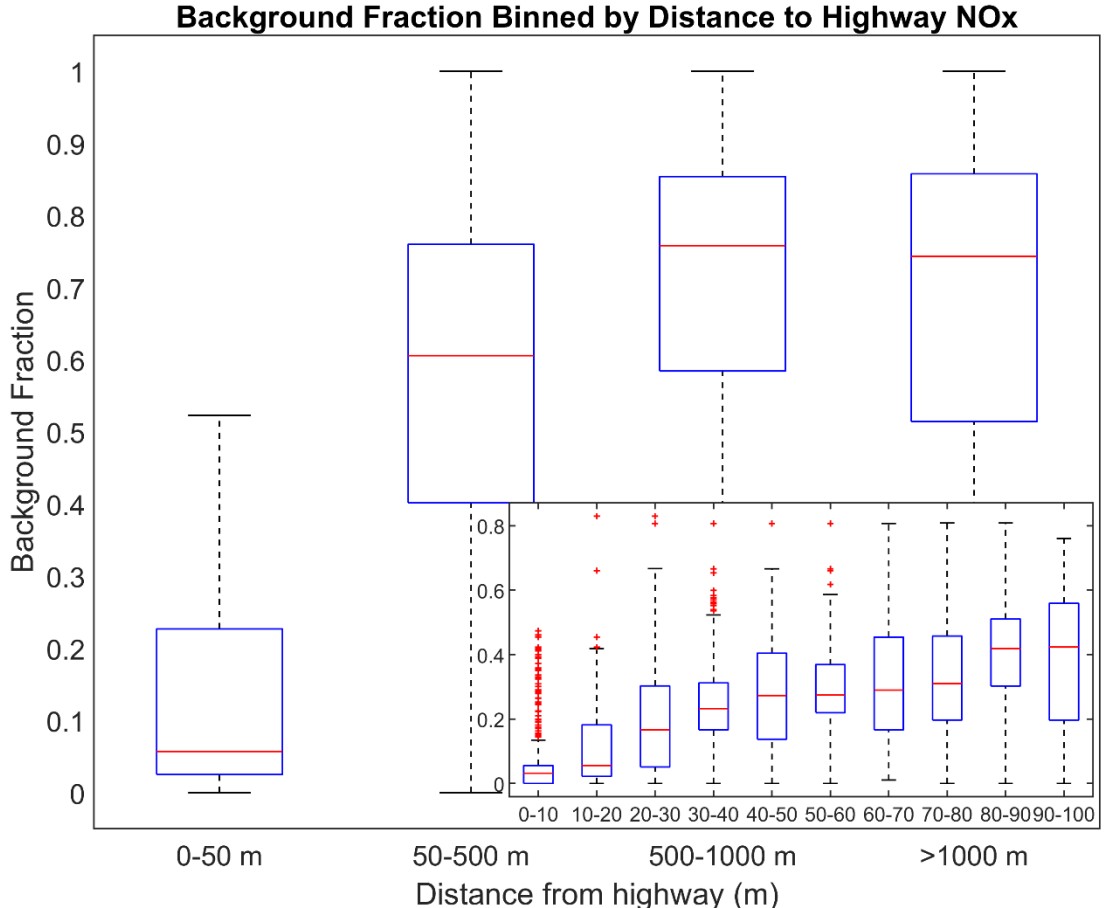

**Figure 6. Boxplots of mapped background NOx fractions binned by distance from highway. The red line represents the median, the top and bottom edges represent the 75th and 25th percentiles, respectively, and the whiskers extend to the most extreme data points not considered outliers.**

## 4.2 Comparison of Source Contribution Maps Using Different Background Removal Techniques

As an illustration of the importance of carefully considering techniques for background quantification and removal and to put

SIBaR calculations in context, we compare the source contribution maps generated using SIBaR to the Apte and Brantley techniques. We zoom in on the Ship Channel quadrant for ease of comparison in Figure 7. We refer the reader to Figures S4-S12 in the Supplement to see maps for all other areas in the mobile monitoring campaign for both pollutants. The average NOx background predicted by the Apte, Brantley, and SIBaR techniques are 15.25 ppb, 11.58 ppb, and 11.14 ppb respectively.




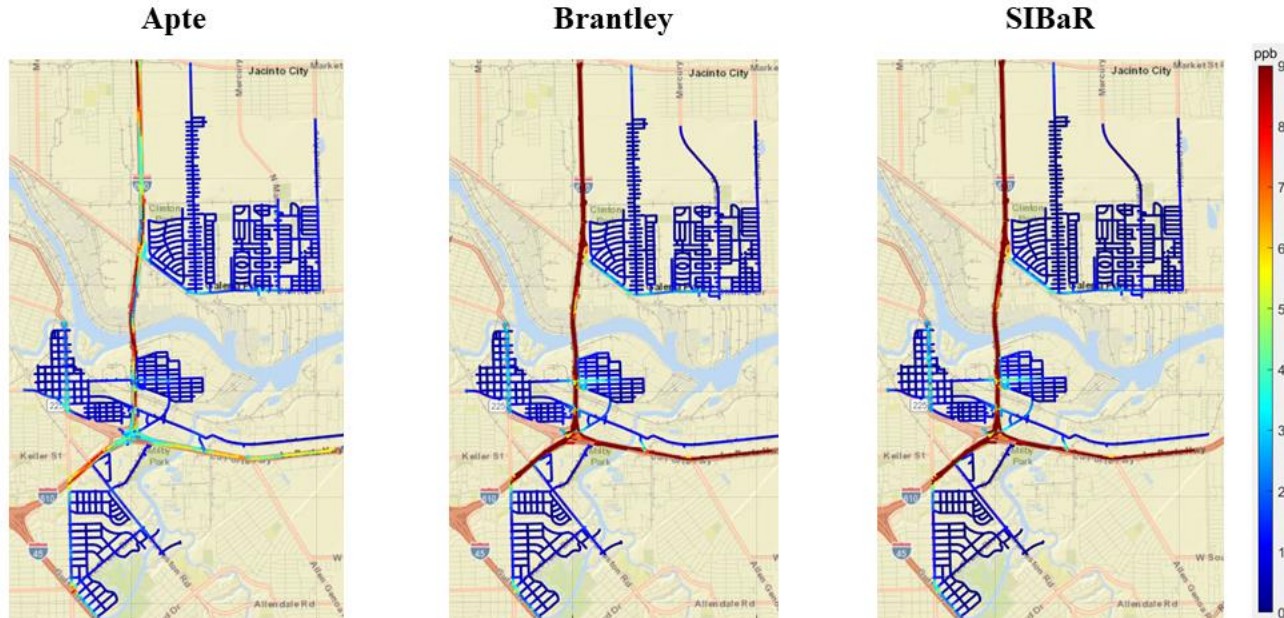

**Figure 7. Comparison of source contributions derived using different techniques in the Ship Channel Quadrant. Source contributions were aggregated according to the methods described in Section 2.4. Basemap generated by Matlab geobasemap 'streets' and is hosted by ESRI (Sources: Esri, DeLorme, HERE, USGS, Intermap, iPC, NRCAN, Esri Japan, METI, Esri China (Hong Kong), Esri (Thailand), MapmyIndia, Tomtom)**

Figure 7 shows that the source contributions derived using the Apte technique are lower on highway compared to the source contributions derived using SIBaR and the Brantley techniques. The Brantley and SIBaR techniques both find higher source contributions on road segments with elevated NO and $NO_2$ concentrations found in Miller et al. (2020) compared to the Apte technique. We hypothesize this occurs due to the smaller time window utilized in the Apte technique. The GSV vehicles would

often sit in traffic on highways for extended periods of time, making a two-minute time window unsuitable for describing source durations during those time periods. While the two-minute assumption would be better suited for situations in which the car was exposed to source durations within that time interval (which is often the case in the Apte study), it would not be for source durations of a larger time interval, highlighting the challenges in assuming a static time window for extensive mobile monitoring campaigns with varying source durations.


We plot road segment median source contributions derived by Apte and Brantley algorithms against the road segment median concentrations derived by SIBaR and present the results for $NO_x$ in Figure 8. Additionally, we plot lines of best fit derived using ordinary least squares (OLS). The bottom panel plot in Figure 8 illustrates that SIBaR derives higher source contributions medians than the Apte technique, largely driven by differences in highway road segment medians. The line of best fit slope

determined using OLS suggests that, on average, SIBaR median source contributions are ~45% higher than Apte median source contributions. The top panel scatter plot between Brantley and SIBaR road segment medians indicates much closer agreement





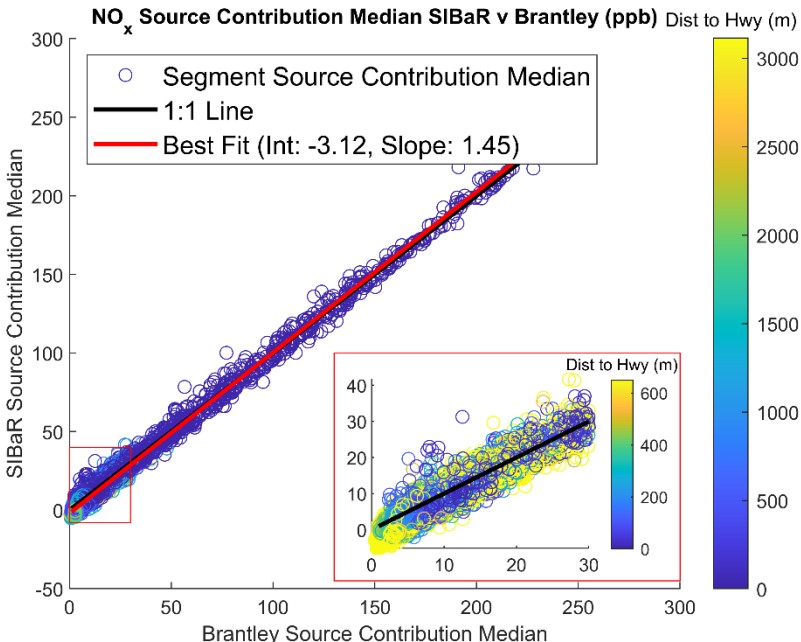

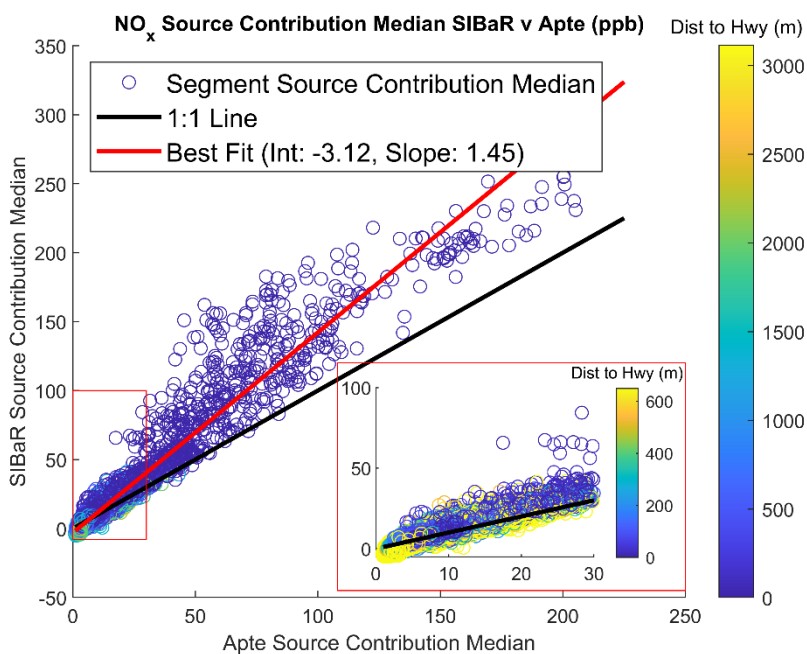

**Figure 8. Scatterplots of road segment median source contributions predicted by two different techniques (designated by "Apte" and "Brantley") against their corresponding SIBaR median source contributions for NOₓ.**

between the two techniques, with SIBaR estimating source contribution medians 2% higher than Brantley source contribution medians. Data for $CO_2$ are shown in Figures S13 and S14.




In addition to plotting the source contribution median, we also plot the source contribution inter quartile range for each road

segment against each other for the different techniques and present them in Figures S15-S18. There are subtle differences in interquartile range between SIBaR and the Brantley technique for both $NO_x$ and $CO_2$, suggesting that different source influences are captured on different days. However, these differences could also be attributed to differences in flexibility between SIBaR and Brantley such that SIBaR consistently predicts lower and more negative source contributions compared to the Brantley technique.

**5 Concluding Remarks**

We illustrate that SIBaR provides a defensible mechanism to quantify and remove background from air pollution monitoring data time series. Most notably, SIBaR does not rely on a static time window assumption to determine source impacts. This time window can have significant impact on the derived source contributions, as exhibited by the discrepancies between the Apte method and SIBaR shown in Section 4.2.

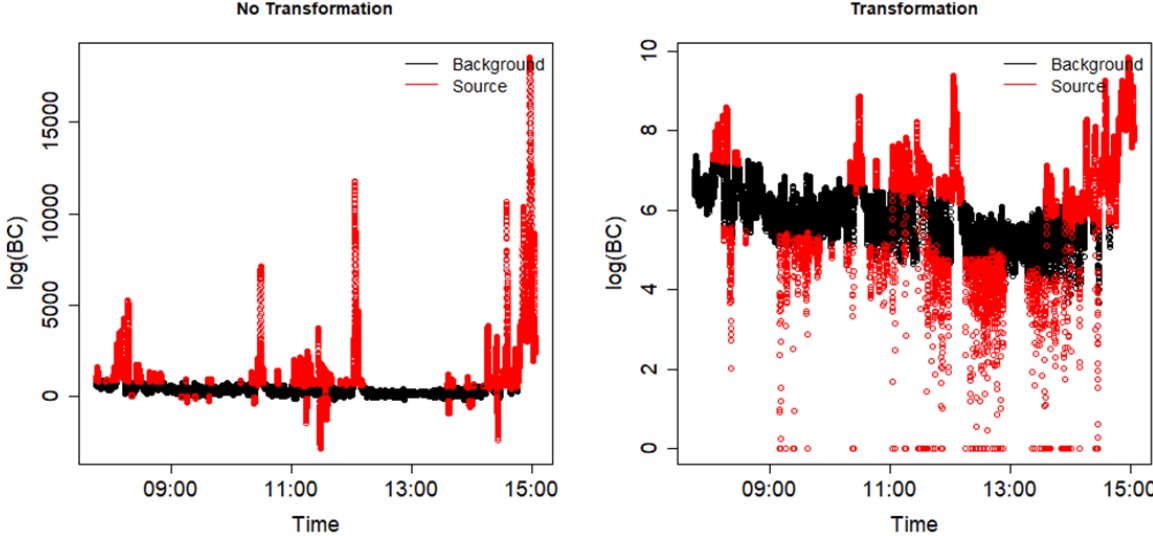

**Figure 9.  Comparison of SIBaR state designations for log-transformed versus non transformed BC data for a day in the Houston mobile monitoring campaign.**

Despite SIBaR's rigor and advancements relative to previously published methods, our approach needs improvement. The analysis is sensitive to noise present within the time series, and smoothing and applying a log transformation does not necessarily eliminate problems associated with this noise. For example, Figure 9 exhibits a side-by-side comparison of SIBaR state predictions for non-transformed BC data and transformed BC data. Due to the noise in the time series, SIBaR is unable to generate a clean partition between background designated points and non-background ones. This sensitivity rests on the



data's ability to be separated into two lognormal distributions. Taking the log of BC data in this case seems to exacerbate problems with skewness in the data distributions.

Problems arise not only with instrument noise and applicability of lognormal distributions to describe data but also with the assumption of a linear time dependence. It is unrealistic to expect background air pollution to exhibit linear behavior, especially

as time series duration extends (Luke et al., 2010). While the linearity assumption seems to be acceptable for time series of several hours of data, problems will most likely arise on time series of data by day or when time series are impacted by abrupt meteorological changes. Future work should incorporate assumptions of non-linear behavior into analysis. Several studies have been published showing the applicability HMMs to covariates expressed as splines (Langrock et al., 2015, 2018). However, trade-offs between computational time and precision would need to be considered. In its current iteration, SIBaR

takes 2.5 hours to model background for millions of data points. The Brantley technique, in contrast, takes several minutes.

Despite these shortcomings, SIBaR holds promise as a framework to quantify and remove background from air pollution monitoring time series. In its current state, it appears inferior to the Brantley technique based solely on computation time. However, problems with SIBaR seem more tied in with computational constraints than with its underlying theory. The SIBaR

partitioning step captures transient behavior between background and non-background quite well, as the diagnostic results of Section 3.1 and the maps in Section 4.1 indicate. In addition to addressing other issues highlighted here, future work should focus on methods to reduce its computational time to make its use more justifiable.

*Code and Data Availability.* Both the code and data are available on request.


*Author Contributions.* BA developed, wrote, and tested the method in R with critical input and scientific guidance from RG and KE. RG supervised the project and provided feedback on significance of method's results. BA wrote the manuscript. All authors contributed to the editing and review of this manuscript.

*Competing interests.* The authors declare that there is no conflict of interest.

*Acknowledgements.* The authors gratefully acknowledge the support of NIEHS (grant #R01ES028819-01). We thank Halley Brantley for the provision of data and comments concerning results in Section 3.1 of the manuscript. We also appreciate support from the Environmental Defense Fund for the collection and provision of mobile data.



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
