# Peer review of "SIBaR: A New Method for Background Quantification and Removal from Mobile Air Pollution Measurements"

_Atmospheric Measurement Techniques, 2021_

## Author Response (AR1)

**Opening remarks**

Please note that all author comments/responses appear in blue throughout this document.

We would like to open by thanking both reviewers for their very thoughtful and thorough reviews. Our response to their calls for major revisions has greatly improved our manuscript. We describe these major changes here and subsequently respond to each reviewer's comment.

The major changes made were the following:

- An implementation into the SIBaR routine to recognize fits such as the ones generated in Figure S1 of the previous manuscript which were a major point of concern. The implementation of this routine should allay the first reviewer's concerns about the susceptibility of this technique's results to outliers and provides more accurate results. The implementation of this routine also removes the necessity of smoothing the data, which addresses the concerns of the second reviewer, who pointed out that the smoothing undercuts the data-only aspect and complicates interpretation of the mapped results.
- We have implemented a new spline fit which fits SIBaR partitioned background points day-by-day rather than fitting all background partitioned points at once on a 2D time scale. The spline fit we've implemented is the same structure as the spline fit used in the Brantley technique – a natural spline with the degrees of freedom equal to the number of hours in the time series. What differentiates our technique from the Brantley approach is our choice to fit to the mean of the background partitioned time series, rather than focusing on a 10th quantile regression of all observations within the time series. Implementing a similar spline structure allows for a more direct comparison the Brantley technique, which is advantageous for this manuscript. Additionally, the implementation of this fit addresses the first reviewer's concerns about the background signals presented in Figure 4 of the original manuscript.
- We have struck out the entirety of Section 3.2 in light of comments made by the second reviewer. We agree that it is inappropriate to describe the monitor as an urban background site (based on its elevation) and then argue that what drives discrepancies between the SIBaR background predictions and the monitor averages are source influences. We suspect that the monitor picks up source influences based on the example time series provided below, which are subsets of the total data compared. In these time series, the monitor registers 5 minute averages of $NO_x$ that are 5-10x than what has been previously published at the site (Luke et al., 2010). Since we have no other observations in place (such as visual/written evidence of source impacts occurring at these specific times) nor additional measurements that could provide information regarding boundary layer conditions (such as ones provided by a ceilometer), we have decided it is most appropriate to strike the entire section.

[Figure]

2018-02-22

2018-03-13

[Figure]

- We have edited our manuscript to make our argument in support of SIBaR as the following. The SIBaR partitioning step successfully partitions background designated observations. We illustrate this in how the partitioning step successfully partitions observations in an external dataset previously marked as background from non-background. Additionally, we are confident the technique successfully partitions background observations because the maps of the fraction of observations designated as background display high values of these fractions in residential areas, with exceptions happening in locations that have been previously published as hotspots. Since the partitioning step successfully decodes observations as background from non-background, an appropriate spline fitting technique that fits to this subset of observations will generate a background estimate that offers advantages over other background estimates that use the entire dataset, especially over ones that rely on a static time window. We illustrate this concept in panel (a) of Figure 8 in the revised manuscript, which shows both the Brantley and Apte techniques over fitting to the data in the early morning hours while the SIBaR technique does not. SIBaR's shortcomings are noted in the bottom panel, which illustrates an example of its background predictions being wildly extrapolated in the early morning hours due to the lack of decoded background states in those early morning hours. We believe this clarifies what advantages the SIBaR technique offers, which was a concern raised by the second reviewer.
- Finally, we have made the following technical corrections that go beyond what the reviewers pointed out:

- We have detected 365 (0.004% of the dataset) measurements of NO that were above the reported dynamic range of the instrument. We have taken these measurements out of the dataset.
- The mapped source contributions derived using the Apte technique displayed in Figure 7 of the original manuscript were of $CO_2$ and not $NO_x$. We have updated the Figure accordingly.
- Figures 5 and 6 in the original manuscript displayed road segments that had fewer than the minimum of 15 passes required to be mapped. We have removed those segments accordingly.
- The maps of median source contributions (displayed in Figures 7,S4-S12) derived using the Apte and Brantley techniques subtracted background from smoothed 30s $NO_x$ and $CO_2$ measurements rather than the original measurements (which were used to generate each background signal). The source contribution medians derived using each technique have been updated accordingly.

**Response to Referee #1**

We would like to thank this referee for providing a very thorough review.

The authors present a new technique for estimating and removing background concentrations of air pollutants from mobile measurement data. A major benefit of this method is that it does not depend on a static time window in estimating background concentrations and instead uses a time-varying approach. The authors compare the results using their developed SIBaR method to two previously-published methods as well as datasets from a stationary site and data collected in a prior mobile measurement campaign. The authors conclude that the SIBaR method results in background estimates similar to those obtained when a previously published technique is used on the same dataset.

Overall my suggestion is to reconsider the paper after major edits have been applied. My two specific points of concern are in regard to the susceptibility of the SIBaR method to outliers and to the high modeled background values shown in the time series in Figure 4. First, the strong dependence of the SIBaR technique on the smoothing time interval is concerning. Looking at Figure S1, the background designations vary wildly and do not make much sense when the data is not smoothed or when 10 second smoothing is applied. Figure S1 also shows that the background designations (in ppm) are much higher than source designations, which is worrisome. If this is not the case then the technique and figure have not been sufficiently explained. Overall the current technique appears too susceptible to time series outliers.

We share the reviewer's concerns regarding the HMM's susceptibility to outliers. The sensitivity of HMMs whose response distributions are modeled as Gaussian to outliers has been noted in the literature (Svensén and Bishop, 2005; Chatzis and Varvarigou, 2007). While those authors explored using heavy-tailed student-t distributions to make their models more robust to outliers, we found more success implementing a routine that recognizes instances of misclassification, breaks the time series in half, performs the partitioning on each separate half, and does so recursively until a reasonable categorization is returned. We describe this routine in Section 2.2 of the revised manuscript.

We also feel that concerns about outliers are allayed by examining how the technique performs across all time series in our campaign. We revised our manuscript to include and emphasize metrics of its success.

Secondly, the SIBaR-modeled background concentrations in the example time series shown in Figure 4 are much too high compared to the other 2 techniques applied to same dataset. The SIBaR-calculated backgrounds in Figure 4 are higher than most of the total ambient measurements, which doesn't make any sense, and the calculated background values do not show enough temporal variability in the background over the course of the day. This Figure is meant to present a single day as an example, however it shows that the SIBaR technique may not be very accurate in modeling background concentrations.

See the major changes above. We have changed how the spline is fit which addresses these concerns.

In addition to these concerns, the authors did not sufficiently address the observations that are an essential part of this research. The paper needs to provide more details on the observations such as the accuracy, precision, limit of detection, etc. of each instrument or measurement technique to provide the reader with some idea of the uncertainty associated with measurements of each pollutant. The reader also does not know the dates or seasons of the measurements, which are vital parts to list for any measurement. This is important because air pollutants such as CO2 which exhibit a seasonal cycle and therefore observations in January cannot simply be compared to those in July. For Table S2, a third column in the table listing the measurement techniques for each of the analyzers would be great.

We have added bias, precision, and MDL estimates to Table S2 along with the corresponding measurement technique. We have added the following text describing when the measurements were taken to section 2.1:

"Measurements were conducted over a 9 month period spanning July 2017 to March 2018. Sampling primarily took place between 7:00 and 16:00 local standard time (Miller et al., 2020) in a variety of census tracts across metropolitan Houston."

Additionally, we added the following paragraph describing the bias, precision, and minimum detection limit for each instrument.

"Bias, precision, and the minimum detection limit (MDL) for each instrument are provided in Table S2. Details concerning the calculation of each parameter for each instrument are given elsewhere (Miller et al., 2020). In brief, the bias for the T200 NO Analyzer and T500U $NO_2$ Analyzer were calculated from gas calibration checks performed every 2 weeks at the start of the study period and every month towards the end of the study period, since the checks routinely showed bias < ±10%. The bias for the Li-COR was determined from a gas phase calibration before the start of the study to match the manufacturer reported value. Precision values for the T200 and T500U were calculated as the standard deviation time series zero composed of zeroing periods taken throughout the entire campaign. Minimum detection limits for the T200 and T500U

were determined from the mean of zeroing period time series + 3σ of that zeroing time series. The minimum detection limit and precision of the Li-COR were not considered due to taking measurements at a consistently elevated global background and the latter manufacturer's reported value having a miniscule effect on the overall uncertainty of the measurement. For the purposes of this work, we perform no MDL substitution, as MDL substitution would censor the underlying modelled background probability distribution."

Below are general comments regarding the figures.

Figures 1, 2, 3, 4, 6, 8: The figures do not "stand alone" if read without the rest of the paper. Even so, some of the figures are not easy to understand even with the text. The captions are very short and need more information such as the dates, locations, and temporal resolution of the measurements, a brief description of the modeling techniques used, etc. Some other suggestions are to take the titles out of the figures and put the information into the figure captions instead. For figures with multiple subplots, designate one as "a" and the other as "b". For the x-axis, specify the time zones (local time, UTC, etc) on your plots. If a legend is provided, make sure it includes units.

Figure 3 specifically: Please check your units for CO. Thousands of ppm is VERY high. I'm not sure if these measurements were taken over a single day or if they were averaged at each time-of-day across multiple days of measurements.

Figure 4 specifically: If I'm understanding this figure correctly, then it appears that SIBaR overestimates the background concentrations by a lot, and this is rather worrisome. The background concentrations estimated by Brantley and Apte appear much more reasonable, although the Apte technique may have too high of a temporal variability.

Figure 8 specifically: Check your y-intercept and slope values in the two plots. They are the same values but the bottom best-fit line should be much different. Please explain the subplot within the figures in the caption. The circles are rather large and hide some of the data behind them. Perhaps consider using small points as markers instead.

We have updated both figures and captions to be less confusing to the reader. Additionally, we have addressed all changes requested in the technical comments below.

Below are specific technical comments to consider for the manuscript.

Specify "Houston, TX" the first time it is mentioned on line 17 (don't need to repeat "TX" on line 19).

Line 31: "Air pollutant concentrations" in place of "pollution concentrations"

Line 32: "ambient background levels" in place of "background"

Line 36: "to determine the background" instead of "to determine background"

Line 47: "background concentrations" instead of just "background"

Line 47: State which traffic-related air pollutants you're looking at here.

Lines 66-67: Need to state the temporal resolution of the data you collected (hourly, minutely, secondly, etc). Also provide the year and months of the campaign. This is especially important for emissions that vary seasonally, such as $CO_2$.

Line 72: "neighborhoods" and "highways"

Lines 75-76: Only need to list the pollutants you did use in your study, not ones that weren't used.

Line 81: "for each GSV" car

Line 130: Please define what variable "k" represents.

Line 130: "day-by-day"

Line 134: "NOx concentrations"

You mention that the RMSE equation is provided by equation (10), but you need to state this equation after the first mention of it here.

Line 139: Include the units for the RMSE of 0.01 (ppb?)

Line 142: "Mobile monitoring dataset?"

Line 143: "Collecting measurements"

Line 143: Define CO at the first mention.

Line 147: State which measurements you're comparing. Carbon monoxide?

Line 151: Be consistent with writing out numbers. In the abstract you wrote "70 m". Either replace "seventy meters" here with "70 m", or replace "70 m" in the abstract.

Line 152: Replace "it" with something like "this site". Also, specify if you mean "Houston background NOx concentrations" or emissions.

Lines 154 and 155: Again, be consistent with spelling out fifth or 5th (similar to Table 1) and tenth.

Line 159: Time series of what pollutants?

Line 168: "…the mean of measurements collected as the GSV car drove past a road segment…"

Lines 169-172: Confusing statement. Please revise.

Line 188: Change "data set" to "dataset".

Lines 189-190: Unclear with what you mean by this sentence.

Line 195: Specify the dates of your comparison.

Line 198: Don't need to redefine RMSE.

Table 2: Need to include more detail in the table caption, including the dates of the measurement periods, the locations, etc. Are the RMSE and MAE values for the entire dataset or an average or median of daily-calculated RMSE and MAE values? Need to state this.

Line 218: "outperforms" in place of "out performs"

Lines 218-219: This is the first time that the reader is introduced to how many days were considered in the analysis. Please be sure to include this information in the methodology with details on the exact dates of measurements.

Figure 4: Place Figure 4 after the paragraph that first mentions it (after line 230).

Lines 223-225: Confusing as written.

Lines 227-228: Please rewrite the second "local pollution influences" with another term to not repeat the same wording twice in the sentence.

Figure 5: Move to the end of line 238 where Figure 5 is first mentioned. Good caption!

Line 267: Please specify which pollutants.

Line 272: "highways" in place of "highway"

Line 283: Add "regression" after OLS

Figure 9: Move to below line 306, after the first mention of the figure

We would like to thank this referee for their thorough review.

**Response to Referee #2**

This paper presents a novel method to quantify and remove background signals from air pollution data, relevant to the processing and interpretation of mobile monitoring measurements. This has the potential to be a significant methodological advance relevant to a broad body of mobile monitoring studies. However, I find that the paper in its current state does not clearly demonstrate that the novel method improves upon prior techniques, nor does it fully justify the additional statistical complexity inherent in this new method. I would suggest significant additions to this analysis to more carefully interpret the output of the HMM, consider the physical/mechanistic interpretation of the signal classification, and

evaluate how the conclusions drawn from mobile monitoring data may be altered by the removal of background signal.

I find two significant flaws in the current analysis. The first is simply that the prior methodology by Brantley et al. (2019) seems to outperform the proposed SIBaR in representing background concentrations and also seems to produce very similar spatial results, leaving the reader questioning the purpose of this more conceptually complex and computationally intensive method. SIBaR has the attractive quality that it provides data-driven, variable time windows in its signal classification scheme, a method that performs well compared with manually classified data (Section 3.1) potentially making it a method that can be adapted to other data series and settings. However, that data-driven quality is somewhat undercut by its sensitivity to the initial smoothing of the input time series. It would be useful for the reader if the authors could point to use cases where the Brantley method may result in significant misinterpretation or misclassification of the data while the SIBaR method performs more favorably. I understand that the comparison against the elevated fixed site monitor may have been an attempt to do so, but I believe there is a fairly reasonable hypothetical explanation for why this failed, which I address in the following paragraph. Are there other ways that SIBaR is more replicable, portable, or robust that provide it an advantage over the Brantley method?

We agree that in generating median concentrations incorporating a substantial amount of measurements that SIBaR offers little advantage over the Brantley technique. However, in applications which involve more precise characterization of the upper portions of the cumulative probability distribution, such as in the identification of elevated 90th percentiles previously published in Miller et al. (Miller et al., 2020), using SIBaR over Brantley or vice versa would have an impact on the analysis. Since SIBaR fits to a subset of the time series, rather than the entire series, we believe it could detect source influences that the Brantley technique could fail to account for. We attempt to show this in Figure 8 of the revised manuscript.

We'd also argue that the decoded states the technique returns are useful. We think HMMs provide excellent unsupervised learning abilities that incorporate time series structure that other popular unsupervised learning techniques, such as k-means clustering, are unable to offer.

The second flaw with this analysis is conceptual and relates to the division of mobile data into two distinct modes of background (defined in the introduction as "measured air pollution independent of local source influences") and non-background/source. I agree with the concept that within mobile measurements there is a hypothetical pollution signal that is time-variant but spatially invariant, and this signal should match concentrations at a semi-remote background monitoring site. However, following the framework described by Shairsingh et al. (2018) (included in this manuscript's references), a mobile measurement represents the superposition of several time- and spatially-variant patterns including (a) this spatially-invariant hourly/daily background, (b) spatially variant/neighborhood elevations, and (c) isolated spikes caused by localized and/or transient emission plumes. Visually, the data classified by SIBaR as "background" appear more similar to (b) type signals than (a), further underscored by how much higher the SIBaR signal appears compared to the Brantley signal in Figure 4. This is not necessarily a failing of the SIBaR method! It is useful

to distinguish between (b) and (c) signals. However, an evaluation of SIBaR results against methods for isolating the (a) signal may sell the method short. There may be a rich array of conclusions that could be drawn by looking at characteristics of the "background" vs. source signals, grouping by neighborhood and considering how parameters of the distributions may vary. I was also curious to know whether any useful information was captured by the time covariate included in the HMM (Line 97-98).

We agree that theoretically there is a superimposition of different distance scales that would not be captured in this framework. We think that SIBaR predicts consistently higher background concentrations compared to the Brantley technique because it is averaging background contributions from several different distance scales. We believe that a background concept incorporating 2 states is still useful even if it is an average of different distance scales, and the inclusion of 3 or more states presents technical and practical difficulties that could be addressed in a future work.

I would suggest that the authors consider the conceptual reasons why the Brantley background removal method outperformed SIBaR in the current evaluation frameworks and whether SIBaR provides value if it is evaluated in a different framework.

We address this in the major changes above and edited the manuscript to emphasize the SIBaR technique's advantages.

 **Specific comments:**

1. This is a nit-pick but applying a temporal smoothing of 30 seconds on data collected in-motion results in a spatial smoothing effect that means that the measurements do not quite match the nominal resolution of 50-meter road points (as per discussion in Chambliss et al. 2020, cited in this work). This isn't a crucial problem, but it does mean that the road segment observations presented in Fig 8 and related analysis aren't independent data points. I don't think it warrants restructuring the analysis, but it may be worth mentioning.

We have responded to this comment by removing the smoothing part from the method. Further details are found in the major changes section.

2. It would be nice to see some additional information on the time parameter described on Line 97.

We have added the following text to section 2.2:

"For the purposes of our work, we assume that the probability distributions governing $y_t$ are log normal and parametrize the mean of the response distribution as:

$$\mu_t = \widehat{\beta_0} + \widehat{\beta_1}t$$

where $\mu_t$ is the time-dependent mean of the response, $\hat{\beta}_0$ and $\hat{\beta}_1$ are estimated parameters, and $t$ is time."

3. RE: Line 98, "we assume that the probability distribution governing yt are log normal" – is this assumption justified for the background signal? I can understand why a time series with plume-related peaks would be log-normally distributed but why would we assume a long tail for background measurements?

We agree that theoretically there should be no long tail for background measurements. We have fit time series with a normal distribution for the background instead of a lognormal one and found little difference in the resulting partitions. We have left the background as lognormally distributed for practical considerations.

4. Figure 2: Is there a reason that these figures present the transformed data and not concentrations? For ease of interpretation, it would be useful to show these in units of ppb.

We have struck this figure from the manuscript. The new figure has units in ppb.

5. Lines 244-245: The authors mention source-dominated hot spots (presumably other than roads) but these are not obvious to the reader, absent local context for interpreting the maps. It would be useful to include annotations on map figures if possible.

We have starred several key hotspot locations in Figure 3.

We have made edits which address these technical corrections below, which include striking the entirety of the stationary monitoring comparison.

**Technical corrections:**

1. Line 53: "a way to determine whether measurements were taken in locations representative of background versus locations subject to local influences." Precisely speaking, the method applies to a time series and not a set of locations, so it determines whether measurements were taken during periods representative of background patterns vs. periods of transient plumes or localized elevations.

2. Line 125: "designate **points** as background or source. State assigned **points**"—to me, "points" suggests a location and here you are referring to observations in a time series, so I would prefer the word "observations"

3. Line 151-154 vs. 228-230: The authors contradict themselves in describing the fixed site as background and then walking that description back when interpreting the results. If you believe that it is influenced by transportation emissions patterns it would be appropriate to include that information in the original description.

4. Figure 8: "Best Fit" description is the same in both panels